# Improvement of Phased Antenna Array Applied in Focused Microwave Breast Hyperthermia

**DOI:** 10.3390/s24092682

**Published:** 2024-04-23

**Authors:** Xuanyu Wang, Zijun Xi, Ke Ye, Zheng Gong, Yifan Chen, Xiong Wang

**Affiliations:** 1School of Information Science and Technology, Shanghai Tech University, Shanghai 201210, China; wangxy22022@shanghaitech.edu.cn (X.W.); xizj@shanghaitech.edu.cn (Z.X.); yeke2023@shanghaitech.edu.cn (K.Y.); 2Yangtze Delta Region Institute (Quzhou), University of Electronic Science and Technology of China, Quzhou 324003, China; zheng.gong@csj.uestc.edu.cn; 3School of Life Sciences and Technology, University of Electronic Science and Technology of China, Chengdu 611731, China; yifan.chen@uestc.edu.cn

**Keywords:** antenna array, biomedical antennas, breast cancer, focused microwave breast hyperthermia (FMBH), hyperthermia

## Abstract

Focused microwave breast hyperthermia (FMBH) employs a phased antenna array to perform beamforming that can focus microwave energy at targeted breast tumors. Selective heating of the tumor endows the hyperthermia treatment with high accuracy and low side effects. The effect of FMBH is highly dependent on the applied phased antenna array. This work investigates the effect of polarizations of antenna elements on the microwave-focusing results by simulations. We explore two kinds of antenna arrays with the same number of elements using different digital realistic human breast phantoms. The first array has all the elements’ polarization in the vertical plane of the breast, while the second array has half of the elements’ polarization in the vertical plane and the other half in the transverse plane, i.e., cross polarization. In total, 96 sets of different simulations are performed, and the results show that the second array leads to a better focusing effect in dense breasts than the first array. This work is very meaningful for the potential improvement of the antenna array for FMBH, which is of great significance for the future clinical applications of FMBH. The antenna array with cross polarization can also be applied in microwave imaging and sensing for biomedical applications.

## 1. Introduction

As an emerging cancer treatment technology, hyperthermia has attracted much research interest and can be used as a potential alternative method for breast cancer treatment [1,2,3,4,5]. The hyperthermia technique heats up the tumor and increases the temperature in the tumor to above 42 °C to destroy the tumor cells [6]. It has become the focus of more and more laboratory and clinical research in the past 30 years [7,8,9,10,11,12,13,14]. The current clinically applicable mechanism of microwave hyperthermia is to insert a thin metallic probe into the human breast and directly transfer microwave energy to the tumor, which is an invasive treatment method [12,13,14].

Focused microwave breast hyperthermia (FMBH) is a new technique for noninvasive and effective breast cancer treatment. FMBH requires the use of microwave phased array antennas (or applicators) to generate a focused beam at the tumor, which can enable selective and accurate heating of the tumor [15,16,17,18,19]. Therefore, the undesired burning of surrounding normal tissues can be largely avoided, so as to reduce potential side effects [20,21,22,23,24,25].

Due to their complicated tissue composition, human breasts are usually highly heterogeneous in terms of the dielectric constant and conductivity distribution. An optimization algorithm is commonly needed in FMBH to optimize the excitation phase/amplitude of each individual element of the antenna array [26,27]. In some special scenarios, however, desired focusing of a breast tumor may not be obtainable by a specific antenna array. This means that the microwave energy in some other parts of the breast is also very high, which can induce unwanted damage of normal tissues. Thus, it is meaningful to exploit novel approaches to improve the focusing performance.

To explore new antenna array configurations for FMBH, we conduct a computational study in this work to investigate the effect of polarization of the antenna array on the microwave focusing results. Two antenna arrays, shown in Figure 1, are investigated. Array 1 has all the elements’ polarization within the vertical planes, i.e., the planes passing the *z* axis. Array 2 has half of the elements’ polarization within the vertical plane and the other half of the elements’ polarization within the horizontal plane, i.e., the *xy* plane, which can be considered as cross polarization. We use six digital realistic human breast phantoms to perform simulations to compare the performances of the two arrays. In each breast phantom, we place a tumor at eight different locations to test the generalization of the two arrays, which leads to 48 different simulation models. For each simulation model, both the arrays are tested, and the antenna excitation phase/amplitude for each case is optimized to perform beamforming at the tumor. The simulation results indicate that the two arrays have largely the same focusing effects for fatty breast phantoms, while array 2 outperforms array 1 in establishing focused fields in dense breast phantoms. In other words, the cross-polarized antenna array, array 2, is more suitable for focusing in highly heterogeneous breast phantoms. We also use some modified breast models to further verify this conclusion. This finding is very useful for the future advancement and clinical applications of the FMBH technique. The cross-polarization configuration may also be meaningful for biomedical microwave imaging and sensing for dealing with a highly heterogeneous environment.

As compared to our previous work [27], this work is fundamentally different, since two antenna arrays are studied and the focusing results are compared. To the best of the authors’ knowledge, this is the first work that systematically studies the effects of polarization on FMBH and proposes a cross-polarized antenna array for better focusing performance in dense breasts. This work clearly shows that antenna polarization is a critical factor that needs to be considered in the array design for FMBH. Another novelty of this work is that we apply modified breast models to demonstrate that array 2 performs better in a more heterogeneous breast model.

Based on the authors’ knowledge, there has not been any previous work studying the effect of antenna polarization on the focusing result of an antenna array used in microwave hyperthermia. The antennas in the arrays of traditional FMBH works are generally configured in the same polarization [28,29]. There are some related works investigating the antenna polarization on the power distribution uniformity in biological samples for microwave-induced thermoacoustic imaging [30,31,32]. It is found that using circular polarizations or two orthogonal linear polarizations can obtain a more uniform microwave power distribution in inhomogeneous biological samples than using single linear polarization, which is beneficial for some imaging applications.

## 2. Materials and Methods

### 2.1. Design of Antenna Array

The simulation setup and the two studied antenna arrays are given in Figure 1a,b. Each antenna array or applicator is a bowl-shaped array and composed of 26 patch antennas. Figure 2a displays the geometry and dimensions of the antenna. The antennas are arranged in three layers, with 12 in the top layer, 10 in the middle layer, and 4 in the bottom layer, as labeled in Figure 1c. These antennas are evenly distributed on a bowl-shaped surface around a digital realistic human breast phantom. The radiated field of the antenna is polarized along the *x* axis in Figure 2a. The operating frequency of the antenna is 2.45 GHz [33], which is an ISM frequency suitable for medical applications. It is reported that 2.45 GHz can result in a penetration depth larger than 5 cm in human breasts. We appreciate this comment. Because this work mainly discusses antenna polarization, we do not deploy a “real bowl” to support the antenna array in the simulations. For a practical array, there has to be a bowl that supports the antennas. In our previous work [28], we fabricated a hemispherical-shaped applicator for focused breast hyperthermia that supported an antenna array similar to the ones in this work. The applicator was created using a 3D printing technique with a dielectric constant of 2.7, which does not affect the focusing effect of the antenna array.

In array 1, all the antennas are placed vertically, and the polarizations are within the vertical planes. In array 2, half of the antennas are placed vertically while the other half are placed horizontally, meaning that half of the antennas’ polarizations are within the horizontal plane. In addition, both adjacent antennas’ polarizations are orthogonal to each other in array 2. This work mainly investigates the effect of polarization.

Oil with a dielectric constant of 2.6 is used as the microwave coupling fluid and fills the gap between the antenna array and the breast phantom [34]. The coupling medium can obviously enhance the impedance matching between the antennas and the breast phantom, which improves the microwave transmission efficiency [35]. This is very meaningful since the microwave power efficiency of the antenna array is a crucial figure of merit for FMBH.

The substrate of the patch antennas has a permittivity of 10.2, which can reduce the antenna size [36,37,38]. In a patch antenna, the metal patch and ground plane essentially form a capacitor. The substrate serves as the dielectric material of the capacitor. A bigger dielectric constant of the material can store more charges in a unit area, and a smaller metal patch and ground plane are needed. This can effectively reduce the size of the antenna. If a small substrate dielectric constant is applied, the antenna is generally much bigger [39,40]. The geometry parameters of the antenna are optimized via CST Microwave Studio. The breast phantom is present when conducting the optimization. The simulated return loss of some antennas (labeled in Figure 2b,d) is shown Figure 2c,e. It is seen that the antennas have good impedance matching at 2.45 GHz. The same antenna was fabricated and experimentally tested in our previous work [28], which shows good performance in FMBH.

Generally, more antenna elements offer more degrees of freedom, which can produce better focusing effects for tumors in the breast with complex structures [33]. However, the focusing effect and the overall array size also need to be balanced. This is because a bigger antenna array (antennas distributed on a bigger bowl) may lead to a bigger focusing spot. Thus, using 26 patch antennas is a tradeoff solution.

The realization of the FMBH method requires a multi-channel microwave source with tunable phase and amplitude for each channel. The microwave source can output either continuous wave or pulsed wave signals, which are both suitable for FMBH [15,16]. An ideal focusing condition requires that the microwave power density at the tumor is higher than that of healthy tissues at other locations. In such a situation, the tumor is selectively heated to 42 °C after several minutes of irradiation and malignant cells undergo apoptosis [3,4], while the normal tissues still maintain a normal temperature.

### 2.2. Digital Realistic Human Breast Phantoms

In view of the vast discrepancies among the parameters of the breasts, including sizes, shapes, and tissue compositions, we apply six digital realistic human breast phantoms to compare the focusing effects of the two arrays. As showcased in Figure 3, the six breast phantoms can be classified into four categories [41], namely mostly fatty (class 1), scattered fibroglandular (class 2), heterogeneously dense (class 3), and very dense (class 4). They are defined according to the amount of glandular tissues in the breast [27,42]. The digital breast phantoms are obtained from the website provided in [40]. Each phantom is composed of many voxels with sizes of 0.5 mm × 0.5 mm × 0.5 mm. The dielectric constant and conductivity of each voxel are defined as a function of frequency. The phantom file is converted to the format of “.vox” and imported into CST. Two text files are also imported to CST together with the .vox file, with one containing the information of the material type of each voxel and the other defining the dielectric constant and conductivity of each material. As observed from Figure 3, the class 1 phantom has the least glandular tissues while the class 4 phantom has the most glandular tissues. The first phantom in Figure 3a belongs to class 1, the second phantom in Figure 3b belongs to class 2, the third to fifth phantoms in Figure 3c–e belong to class 3, and the last phantom in Figure 3f belongs to class 4. Since breast phantoms in class 3 are very heterogeneous and focusing on such phantoms maybe more difficult, we investigate 3 breast phantoms to provide more results to justify the obtained conclusions. Due to a limit of space, we do not study more phantoms for classes 1, 2, and 4. Each phantom is tested eight times, and a 1 cm diameter tumor is embedded at a randomly chosen location in the phantom in each case, as depicted in Figure 3. The dielectric constant and conductivity of breast tumors are 65 and 2.5 S/m, respectively, at 2.45 GHz [43]. Thus, there are in total 48 different scenarios to study.

### 2.3. Optimization Algorithm for Microwave Focusing

Due to the need to achieve precise microwave focusing on a tumor in a highly inhomogeneous environment such as human breasts, there is no simple analytical method to calculate the excitation phase/amplitude of the antenna array. An optimization algorithm must be used to perform multiple iterations to obtain the optimal phase/amplitude. In each iteration, the microwave power deposition in the breast phantom is obtained through some monitoring or guidance mechanism. In experiments or clinical scenarios, the power density distribution in the breast can be detected by methods such as MRI [44], ultrasound imaging [45], and thermoacoustic imaging [46]. In the current simulation study, we just need to record the power distribution in the breast phantom in CST software.

In this work, we use an optimization algorithm called differential evolution (DE) [47,48] to implement the searching for the optimal phase/amplitude. This algorithm has been well demonstrated in some previous works [27,28,48]. The particle swarm optimization (PSO) method has an additional weighting factor in its update equation, which is called inertia weight. It controls the impact of the iterative speed of the previous step on the iterative speed of the current step. The bigger this inertia weight, the better the global searching ability. Generally, this inertia weight is set to 1, which can gradually reduce the global searching ability as the iteration evolves. This means that it is likely that the finally obtained solution is just a local optimal solution rather than the global optimal solution. On the contrary, the DE algorithm relies on crossover and mutation operations that can efficiently avoid getting a local optimal. Thus, the DE algorithm performs better than the PSO algorithm. It has been found that the excitation phase imposes a much larger impact on the focusing effect than the excitation amplitude. Therefore, only the phase is optimized in this work and all the amplitudes are set to uniform.

In order to realize the DE algorithm, a fitness function needs to be defined as the evaluation criterion for each iteration step. We adopt the fitness function as
(1)$$Q_{\mathrm{ratio}}=\frac{Q_{\mathrm{tumor}}}{Q_{\mathrm{healthy}}}$$
where the power density *Q* is calculated by
(2)$$Q=\sigma {\left|{\overset{⃑}{E}}_{\mathrm{tot}}\right|}^{2}=\sigma {\left|\sum_{i=1}^{26}e^{j\varphi_{i}}{\overset{⃑}{E}}_{i}\right|}^{2}=\sigma {\left|\sum_{i=1}^{26}e^{j\varphi_{i}}{\overset{⃑}{E}}_{ix}+\sum_{i=1}^{26}e^{j\varphi_{i}}{\overset{⃑}{E}}_{iy}+\sum_{i=1}^{26}e^{j\varphi_{i}}{\overset{⃑}{E}}_{iz}\right|}^{2}\;=\sigma \left({\left|\sum_{i=1}^{26}e^{j\varphi_{i}}{\overset{⃑}{E}}_{ix}\right|}^{2}+{\left|\sum_{i=1}^{26}e^{j\varphi_{i}}{\overset{⃑}{E}}_{iy}\right|}^{2}+{\left|\sum_{i=1}^{26}e^{j\varphi_{i}}{\overset{⃑}{E}}_{iz}\right|}^{2}\right)$$
where *Q* is the microwave power density in W/m^3^, *σ* is the electrical conductivity of the breast tissues, *φ_i_* denotes the excitation phase of the *i*th antenna, *E_i_* is the electric field in the breast phantom solely contributed by the *i*th antenna, *E_i__x_*, *E_i__y_*, and *E_i__z_* are the three components, and *E*_tot_ is the total electric field (a complex number) in the breast phantom contributed by all the 26 antennas. *Q*_tumor_ is the average microwave power density in the tumor, and *Q*_healthy_ is the maximum microwave power density in healthy tissues.

In each step of the iterative optimization, the phase of each individual antenna is adjusted by the optimization algorithm. The algorithm uses the microwave focusing results (i.e., microwave power distribution) of the previous steps as feedback information for the next iteration to maximize the fitness function in Equation (1). In this way, the value of the fitness function is gradually increased, which means that the hot spots in the healthy tissue region are gradually eliminated. Once the fitness function reaches a desired value, an acceptable focused microwave field is achieved, and the iteration is terminated. This algorithm has been proven to be superior to particle swarm optimization.

### 2.4. Procedure of Computational Study

First, we conduct a microwave simulation of the complex breast phantom by CST software [49]. A total of 26 sets of CST simulations are performed using a breast phantom containing a tumor. In each set of simulation, only one antenna is radiating while the excitation signal of the rest 25 antennas is set to 0. Then, the electric field in the entire breast due to the *i*th radiating antenna is recorded, which is *E_i_* in Equation (2). This process is repeated 26 times, and in total, 26 sets of electric field distributions are obtained.

Second, we employ the DE algorithm to optimize the fitness function and the excitation phase. The initial phase of the antennas can be set to 0 or an arbitrary value. In each iteration, we calculate *Q* by Equation (2) using the excitation phase of all 26 antennas (*φ*_1_ to *φ*_26_) in the current iteration. It should be noted that although the 26 sets of electric fields are obtained individually, they can be used to synthesize the total electric field as the 26 antennas radiate together. It should be noted that *Q* in Equation (2) is not directly calculated by CST. We first use CST to perform 26 sets of simulations to obtain *E_i_* and export the 26 sets of *E_i_* data into MATLAB. Each set of *E_i_* data is composed of three 3D matrices containing the *x*, *y*, and *z* components of *E_i_* in complex numbers, i.e., *E_ix_*, *E_iy_*, and *E_iz_* in Equation (2). Then, we use Equation (2) to calculate *Q* in MATLAB. Then, we use Equation (1) to determine whether an acceptable focus has been achieved. If not, the DE algorithm yields a new set of excitation phases, and the iterative optimization is continued. Generally, *Q*_ratio_ > 1 is a desired result of the optimization, which means the power in the healthy tissues does not exceed that in the tumor. However, such a desired condition may not be obtainable in some cases. Thus, a more practical goal is to optimize *Q*_ratio_ until its value becomes convergent.

In this work, we set the maximum number of iterations to be 100, which is because the *Q*_ratio_ of most of the studied cases can no longer be increased after 100 iterations. In other words, the optimization process becomes convergent, and more iterations are not needed. Then, the optimization process is terminated. The value of the final *Q*_ratio_ and corresponding excitation phase can be obtained [26].

## 3. Results

### 3.1. Focusing Results Using Original Breast Phantoms

We then compare the focusing effects of the two antenna arrays. The simulated microwave power density distributions of all the 96 studied cases are shown in Figure 4, Figure 5, Figure 6, Figure 7, Figure 8 and Figure 9. Each figure has eight sets of subfigures and each set has three subfigures (e.g., Figure 4a,i,q or Figure 4g,o,w) representing the conductivity distribution, power density distribution using array 1, and power distribution using array 2. To be specific, the first and fourth columns of each figure of Figure 4, Figure 5, Figure 6, Figure 7, Figure 8 and Figure 9 display the sagittal plane view of the conductivity distribution of the breast phantom with a tumor, the second and fifth columns give the corresponding power density for array 1, and the third and sixth columns present the corresponding power density for array 2. Figure 4, Figure 5, Figure 6, Figure 7, Figure 8 and Figure 9 are plotted using a hot colorbar. The circular bright spot in the conductivity distribution is the tumor, which has the highest conductivity among all the tissues. Both the antenna arrays can render good microwave-focused fields at the tumor for most of the 48 studied cases since the tumor obtains the highest power among all the breast tissues. Some parts of the skin can also absorb high microwave power due to its high conductivity. But some measures can be taken to cool down the skin to avoid burning it. The basic cooling method is circulating the oil. We can make two holes in the applicator and use two tubes to connect to a pump. We put the tubes in a big tank filled with an ice–water mixture. By controlling the pump, the oil can be circulated in the tubes and effectively cooled down by the ice–water mixture. More details can be found in previous work [38].

### 3.2. Quantitative Comparison

In order to quantitatively evaluate the focusing effect, the ratio of the fitness function *Q*_ratio_ of the two arrays defined in Equation (1) is provided in Figure 10. For most of the cases, *Q*_ratio_ is greater than 0.7. For those conditions with *Q*_ratio_ < 0.7, the maximum power density in the healthy tissue probably occurs around the tumor, which is acceptable for the treatment purpose of FMBH. According to the definition of *Q*_ratio_ in Equation (1), the condition *Q*_ratio_ > 1 indicates that the average microwave power density in the tumor is greater than the maximum microwave power density in healthy breast tissues. This means the tumor can be heated up by the microwave signal with the highest efficiency and reach the highest temperature. Meanwhile, the healthy breast tissues are heated up to a temperature lower than that in the tumor, and undesired burning in healthy tissues can be avoided. Therefore, accurate treatment of the breast tumor is achieved.

For a more intuitive and generalized comparison, the *Q*_ratio_ of the same breast phantom with different tumor locations is averaged to obtain *Q*_avgratio1_ and *Q*_avgratio2_, in which 1 and 2 in the subscripts represent array 1 and array 2, respectively. For example, the *Q*_avgratio1_ for the class 4 breast phantom means the average *Q*_ratio_ of all eight tested tumors in Figure 3f using array 1. Thus, each of the six tested breast phantoms results in one *Q*_avgratio1_ and one *Q*_avgratio2_. Then, a new ratio defined in (3) is obtained.
(3)$$Q_{\mathrm{ratio}21}=\frac{Q_{\mathrm{avgratio}2}}{Q_{\mathrm{avgratio}1}}$$

It should be noted that *Q*_ratio21_ is the major figure of merit of this work. If the *Q*_ratio21_ > 1, array 2 has better microwave focusing performance than array 1. The results of Equation (3) of the six breast phantoms are plotted in Figure 11. It is seen that for the class 1 and class 2 phantoms, if *Q*_ratio21_ ≈ 1, the focusing effects of the two arrays are comparable. It is straightforward that the focusing effect of array 2 is better than that of array 1 for the three class 3 phantoms and the class 4 phantom since the *Q*_ratio21_ > 1.2. The improvement of array 2 in the focusing performance over array 1 is greater than 20%. As a result, we can reach a conclusion that array 2 outperforms array 1 in terms of the focusing effect for FMBH in dense breasts, while the two arrays perform equally well for FMBH in fatty breasts. To be more specific, unwanted hot spots in healthy breast tissues can be better suppressed by array 2 than array 1 for the cases of dense breasts. Furthermore, the advantage of array 2 in terms of microwave focusing is more obvious in denser breasts or breasts with higher inhomogeneity. This implies that the microwave focusing effect can be enhanced to a certain extent by simply using an antenna array embracing orthogonal polarizations without adjusting the antenna structure or increasing the number of antennas. Orthogonal polarization tends to offer more flexibility in microwave focusing and leads to a better focused field, especially in dense breasts.

To achieve good microwave focusing of a tumor in a highly inhomogeneous breast or to maximize *Q*_ratio_ in Equation (1), we not only need to increase the microwave power in the tumor, but also want to reduce the microwave power in potential hot spots in the healthy tissues. Applying cross polarization or polarizations in two orthogonal directions can improve the flexibility and efficiency in reducing the hot spots in healthy tissues. In addition, the propagation of an electromagnetic wave in an inhomogeneous medium is highly related to the wave polarization. For example, it is easier for an x-polarized wave than a y-polarized wave to traverse a medium containing some elongated high-permittivity obstacles along the y direction. This means that orthogonal polarization is more flexible in beamforming in homogeneous media.

### 3.3. Focusing Results Using Modified Breast Models

We perform further studies to compare the two arrays. We use the class 4 phantom with a tumor as the case in Figure 9d and modify the tissue properties in it. As shown in Figure 12a, the original model is named Model 1. The permittivity as well as conductivity of the healthy breast tissues in the phantoms in Figure 12b,c are reduced to 70% and 30%, respectively, of those in Model 1, which are referred to as Model 2 and Model 3. The dielectric property of the tumor is not changed. The three models are investigated by the two arrays and the microwave power distributions after 100 iterations are shown in Figure 12d–i, with Figure 12d–f for array 1 and Figure 12g–i for array 2.

The corresponding *Q*_ratio_ results are shown in Figure 13. It can be observed that as the permittivity and conductivity of the healthy tissues decrease, the *Q*_ratio_ of the two arrays gradually increases, meaning that the focusing effect becomes better. At the same time, the increase in the focusing effect of array 2 is relatively smaller than that of array 1. Moreover, for the densest breast model, Model 1, the focusing effect of array 2 is better than that of array 1. For Model 3 that is nearly a fatty breast, array 1 performs better than array 2. For Model 2, the two arrays lead to largely the same focusing effect. These results further prove that array 2 is more favorable for establishing a good focused microwave field in a dense breast.

This finding is also valuable for microwave imaging or sensing applications based on antenna arrays. For most of the microwave imaging or inverse scattering techniques, antennas are configured with the same polarizations around the sample under test [50,51]. However, exploring imaging applying dual polarization or cross polarization can offer the possibility of extracting more information and acquiring higher image quality of the target. In addition, Figure 13 implies that a cross-polarized antenna array is favorable for imaging highly inhomogeneous objects like dense breasts.

## 4. Discussions and Conclusions

In a practical FMBH system, we can fabricate a holder to accommodate all the antenna elements in Figure 1b. We applied a multi-channel microwave source to excite the antennas with tunable amplitude and phase to perform the beamforming. The multi-channel microwave source can be efficiently controlled by a computer. To enable practical applications of the FMBH technique, a monitoring technique is needed to judge if a desired focused field at the tumor is established. Some measures such as microwave-induced thermoacoustic tomography can be taken to implement the monitoring task. More details can be found in our previous works [28,38,52].

To summarize, we propose a cross-polarized antenna array to improve the microwave focusing in breast hyperthermia. Six real breast phantoms with different densities and different tumor positions were used to conduct numerical simulations to compare the focusing effect of two antenna arrays. The complete process and data comparison were presented in detail. The simulated power density distribution indicates that the arrangement of antenna array 2 can obviously improve the focusing quality in heterogeneously dense and very dense breast phantoms. This computational research is of great significance for improving the performance of FMBH and the microwave imaging/sensing technique.

## Figures and Tables

**Figure 1 sensors-24-02682-f001:**
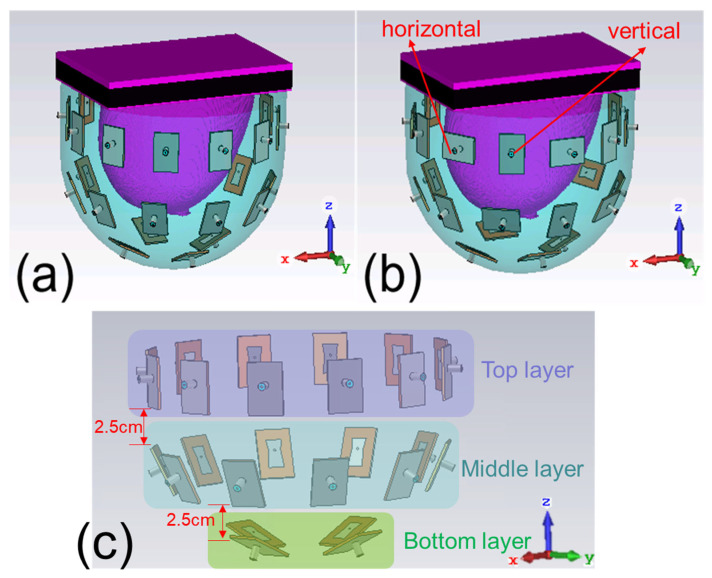
Configurations of the two studied antenna arrays: (**a**) array 1; (**b**) array 2 (cross polarization); (**c**) three layers of antenna arrays. Purple represents the top layer, blue represents the middle layer, and green represents the bottom layer.

**Figure 2 sensors-24-02682-f002:**
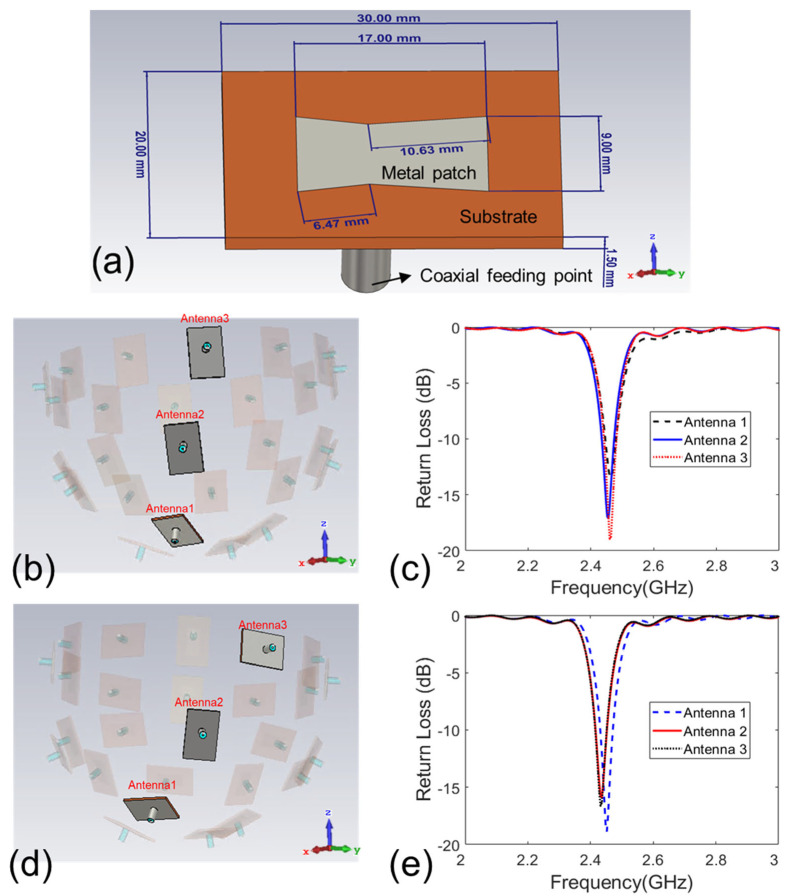
Details of the antenna: (**a**) the structure of the patch antenna; (**b**) three typical antennas reside in the three layers for the result in (**c**); (**c**) simulated return loss of typical three patch antennas in array 1 with the breast phantom present; (**d**) three typical antennas reside in the three layers for the result in (**e**); (**e**) simulated return loss of three typical patch antennas in array 2 with the breast phantom present.

**Figure 3 sensors-24-02682-f003:**
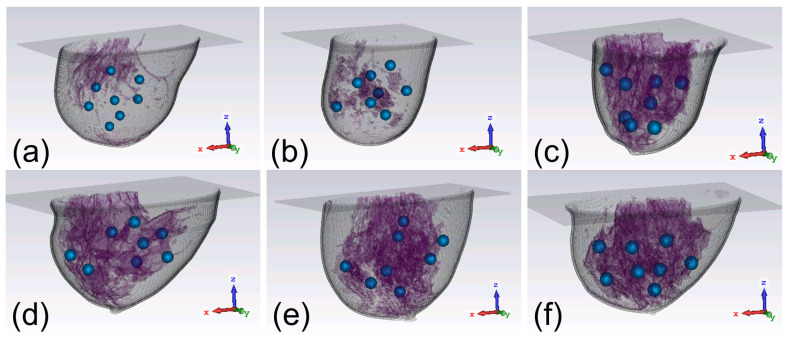
Six breast phantoms with tumors: (**a**) class 1; (**b**) class 2; (**c**) class 3 A; (**d**) class 3 B; (**e**) class 3 C; (**f**) class 4. The small blue balls represent the tumors and the complicated structures in purple are mainly the breast glandular tissues.

**Figure 4 sensors-24-02682-f004:**
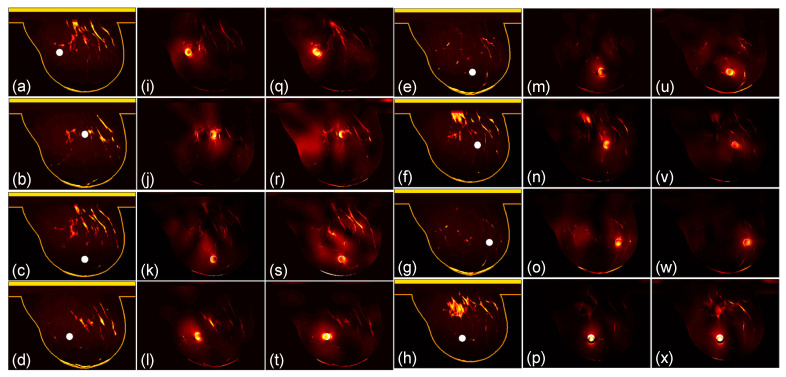
(**a**–**h**) The conductivity distribution of class 1 breast model with 8 different tumor positions; (**i**–**p**) the power density distribution of antenna array 1; (**q**–**x**) the power density distribution of antenna array 2. The brightest circular spot in the conductivity distributions is the tumor. The U-shaped yellow contour line represents the skin layer of the breast phantom.

**Figure 5 sensors-24-02682-f005:**
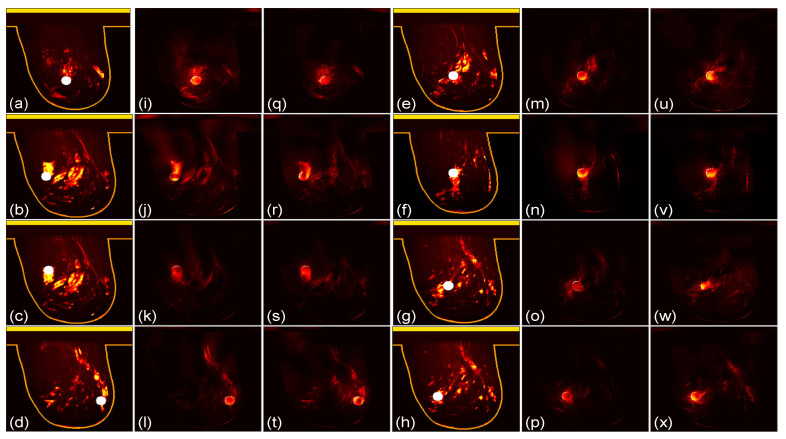
(**a**–**h**) The conductivity distribution of class 2 breast model with 8 different tumor positions; (**i**–**p**) the power density distribution of antenna array 1; (**q**–**x**) the power density distribution of antenna array 2. The brightest circular spot in the conductivity distributions is the tumor. The U-shaped yellow contour line represents the skin layer of the breast phantom.

**Figure 6 sensors-24-02682-f006:**
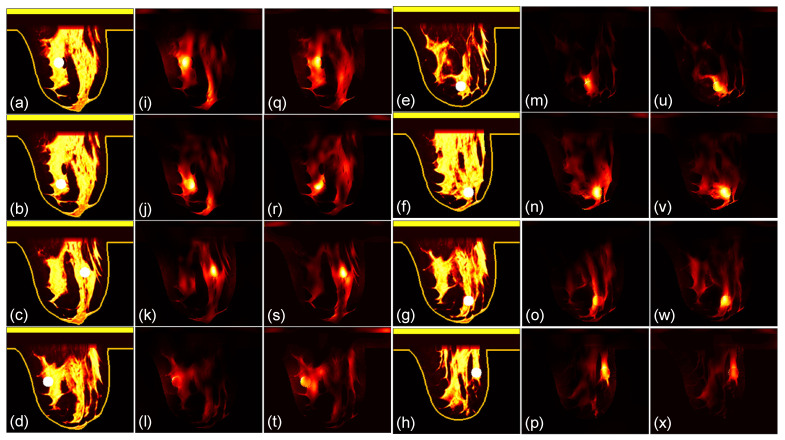
(**a**–**h**) The conductivity distribution of class 3 A breast model with 8 different tumor positions; (**i**–**p**) the power density distribution of antenna array 1; (**q**–**x**) the power density distribution of antenna array 2. The brightest circular spot in the conductivity distributions is the tumor. The U-shaped yellow contour line represents the skin layer of the breast phantom.

**Figure 7 sensors-24-02682-f007:**
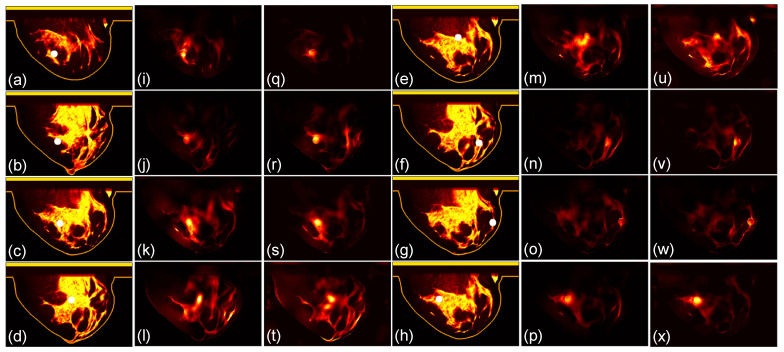
(**a**–**h**) The conductivity distribution of class 3 B breast model with 8 different tumor positions; (**i**–**p**) the power density distribution of antenna array 1; (**q**–**x**) the power density distribution of antenna array 2. The brightest circular spot in the conductivity distributions is the tumor. The U-shaped yellow contour line represents the skin layer of the breast phantom.

**Figure 8 sensors-24-02682-f008:**
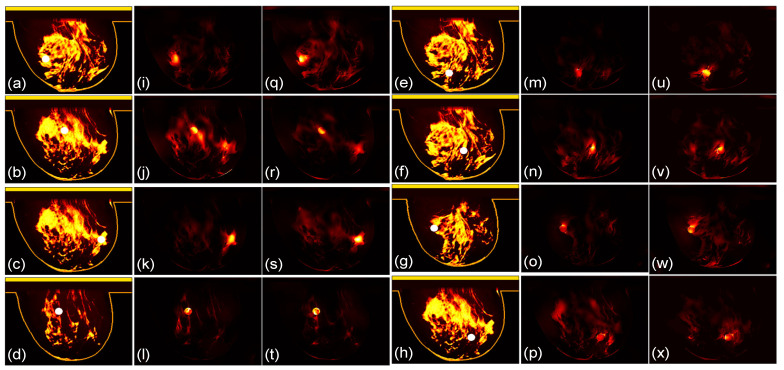
(**a**–**h**) The conductivity distribution of class 3 C breast model with 8 different tumor positions; (**i**–**p**) the power density distribution of antenna array 1; (**q**–**x**) the power density distribution of antenna array 2. The brightest circular spot in the conductivity distributions is the tumor. The U-shaped yellow contour line represents the skin layer of the breast phantom.

**Figure 9 sensors-24-02682-f009:**
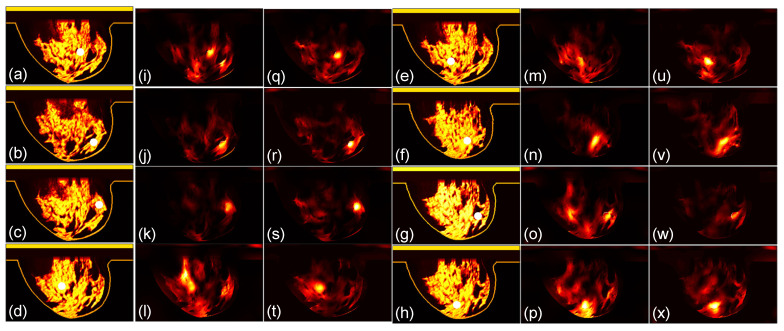
(**a**–**h**) The conductivity distribution of class 4 breast model with 8 different tumor positions; (**i**–**p**) the power density distribution of antenna array 1; (**q**–**x**) the power density distribution of antenna array 2. The brightest circular spot in the conductivity distributions is the tumor. The U-shaped yellow contour line represents the skin layer of the breast phantom.

**Figure 10 sensors-24-02682-f010:**
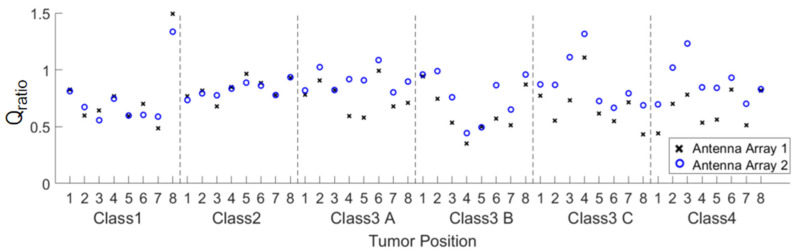
The value of *Q*_ratio_ of the focusing results of all 48 models by the two arrays.

**Figure 11 sensors-24-02682-f011:**
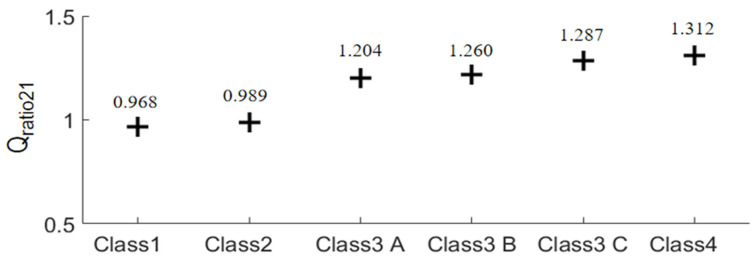
Comparison of the averaged *Q*_ratio21_ of the focusing results of the two arrays in the six breast phantoms.

**Figure 12 sensors-24-02682-f012:**
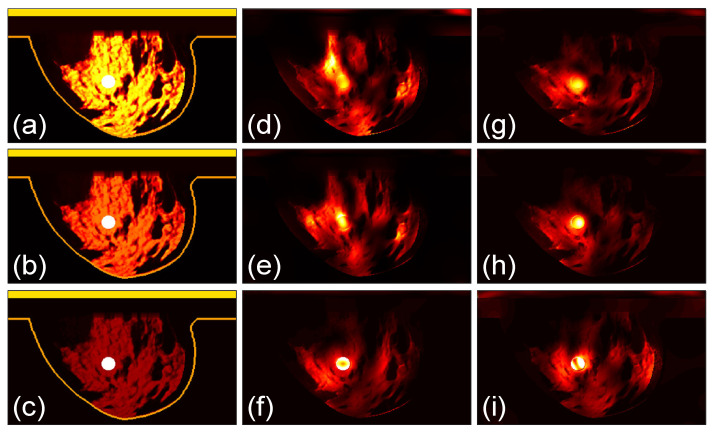
(**a**–**c**) The conductivity distribution of modified breast Model 1, Model 2 and Model 3, which are based on the class 4 phantom in Figure 9d; (**d**–**f**) the power density distribution using antenna array 1; (**g**–**i**) the power density distribution using antenna array 2. The brightest circular spot in the conductivity distributions is the tumor. The U-shaped yellow contour line represents the skin layer of the breast phantom.

**Figure 13 sensors-24-02682-f013:**
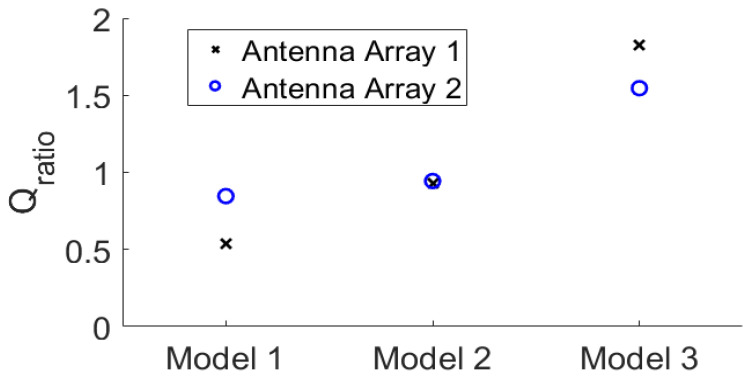
The focusing results using the modified breast models in Figure 12.

## Data Availability

Data are available upon request.

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
