# Peer review of "Improvement of Phased Antenna Array Applied in Focused Microwave Breast Hyperthermia"

_sensors, 2024, doi:10.3390/s24092682_

Round 1

Reviewer 1 Report

Comments and Suggestions for Authors

The authors proposed an improved phase array antenna for focused microwave breast hyperthermia by considering various digital breast phantoms. The research topic is interesting and the manuscript is well organized and written. This reviewer recommends publication of this manuscript in Sensors with the following minor comments.

1. Page 3, Line 3: You described the arrangement of antennas at each layer. Please provide details about the spacing between antennas within each layer.

2. Page 3, Line 5: You mentioned that “These antennas are evenly distributed on a bowl-shaped surface around a digital realistic human breast phantom”. Further explanation is needed, such as the material and dielectric constant of the bowl surrounding the human breast phantom.

3. Fig. 2 (b) and (c):  Fig. 2 (b) and (c) display the return loss of three patch antennas in array 1 and array 2, respectively. An explanation is required regarding to the positions of the antennas 1, 2, and 3 shown in the figures.

4. Page 4, Line 21: The breast phantoms classified into four categories. However, why was class 3 further subdivided into three types? Please address it in detail.

5. Page 4, Line 22: In this paper, the tumor is a key focus; however, specific information on the tumor, such as its dielectric constant and conductivity, is lacking. Please provide these parameters along with a detailed explanation.

6. Page 5, Line 30: You mentioned that “This algorithm has been proven to be superior to the particle swarm optimization”. Can you briefly explain why differential evolution (DE) is considered superior to particle swarm optimization?

7. Page 6, Line 20: You mentioned that “But some measures can be taken to cool down the skin to avoid burning in it”. Please explain in detail how to cool the skin excluding the tumor.

8. Fig. 4, 5, 9: In the previous section, breast phantoms were broadly divided into four categories, with class 3 further subdivided into three types. However, in Fig. 4, 5, and 9, class 1, 2, and 4 are also labeled as class A. Is there a specific reason for this discrepancy?

9. Page 9, Eq. (3): Please explain what means of “Qavgratio1” and “Qavgratio2”.

Reviewer 2 Report

Comments and Suggestions for Authors

I would like to thank the reviewers to bring the submitted mansuscript togther with title, "Improvement of Phased Antenna Array Applied in Focused Microwave Breast Hyperthermia", In order to consider the submitted manuscript, the follwoing comments should be addressed as follows:

1. The introduction does not show relevant or similar work and comparing the contribution of the submitted work.

2. The results and discucsion need to be clarified with more details.

3. The authors present full simulation study for an array , however the mansucript should be updated how the proposed study could be applied practically.

4. The choice of lossy substrate of dielectric constat 10 to reduce size should be justified with details of antenna and its response.

5. Details of breast phantom model is not presented and how different class of breast is modelled in simulations.

6. The authors should emphasis the figure of merit of the system and clarify the defintion and its impact.

Reviewer 3 Report

Comments and Suggestions for Authors

Through simulations, this paper examines how antenna element polarization affects microwave focusing outcomes. The researchers used two distinct computerized realistic hu-man breast phantoms to investigate two types of antenna arrays with an equal number of elements. However, there are some concern need be addressed before considering it for publication

1.       How cross polarized antenna array improve focusing performance?

2.       Proof reading is required. For ex. “The simulation setup and the two studied antenna arrays are given in Figure 1. Each antenna array or applicator is a bowl-shaped array and composed of 26 patch antennas, as shown in Figure 2(a).” Fig 2a is simple patch antenna, there is a mismatch of figure numbers

3.       The antennas are arranged in three layers, with 12 on the top layer, 10 in the middle and 4 on the bottom layer”- this arrangement is not clearly shown in any figure

4.       The final obtained return loss of some antennas is shown in Figures 2(b) and (c).” which antenna performance is shown and why its is chosen. Whether all the antenna resonance will be same or only some antenna resonates at 2.4 GHz. There should more clarity on this segment

5.       What is the reason for choosing Qratio >1? Which should be explained in the manuscript

6.       The impact of cross polarized array compared with linearly polarized can be discussed further with the help of results

7.       How the value of Q is computed form equation 2. Is it calculated directly from CST? The detailed steps in calculations Q values can be added

Comments on the Quality of English Language

Minor editing required

Round 2

Reviewer 3 Report

Comments and Suggestions for Authors

Comments were addressed